# Hyperammonemia in Russia Due to Carbonic Anhydrase VA Deficiency Caused by Homozygous Mutation p.Lys185Lys (c.555G>A) of the *CA5A* Gene

**DOI:** 10.3390/ijms232315026

**Published:** 2022-11-30

**Authors:** Natalia Semenova, Andrey Marakhonov, Maria Ampleeva, Marina Kurkina, Galina Baydakova, Mikhail Skoblov, Natalia Taran, Olga Babak, Ekaterina Shchukina, Tatyana Strokova

**Affiliations:** 1Research Centre for Medical Genetics, Moscow 115522, Russia; 2Independent Clinical Bioinformatics Laboratory, Moscow 123181, Russia; 3Master’s Programme “Data Analysis in Biology and Medicine”, National Research University “Higher School of Economics”, Moscow 101000, Russia; 4Federal Research Centre of Nutrition and Biotechnology, Moscow 115446, Russia; 5Perinatal Center at City Clinical Hospital No. 24, Moscow 127287, Russia

**Keywords:** carbonic anhydrase VA deficiency, *CA5A* gene, hyperammonemia

## Abstract

Hyperammonemia due to carbonic anhydrase VA deficiency (OMIM# 615751) is a rare, life-threatening hereditary disease caused by biallelic mutations in the *CA5A* gene, presenting as encephalopathic hyperammonemia of unexplained origin during the neonatal period and infancy. Here, we present a detailed description of a 5-year-old patient with the homozygous mutation p.Lys185Lys (c.555G>A) in the *CA5A* gene. This variant was previously described by van Karnebeek et al. in 2014 in a boy of Russian origin. We found a high frequency of carriers of this mutation in Russia; 1:213, which is 7 times higher than the expected frequency calculated based on data on Western European populations. Thus, targeted testing for the mutation p.Lys185Lys (c.555G>A) in the *CA5A* gene should be useful for early detection by selective screening in neonatal intensive care units.

## 1. Introduction

Carbonic anhydrase VA (CAVA) is an enzyme that supplies bicarbonate as a substrate to four mitochondrial enzymes: pyruvate carboxylase (PC), carbamoyl phosphate synthetase (CPS1), 1,3-methylcrotonyl-CoA carboxylase 1 (3MCC), and propionyl-CoA carboxylase (PCC). Biochemical evaluation shows multiple metabolic abnormalities and evidence of impaired provision of bicarbonate to essential mitochondrial enzymes. Carbonic anhydrase VA deficiency (OMIM# 615751) is an autosomal recessive inborn error of metabolism, characterized clinically by acute onset of hyperammonemic encephalopathy (feeding intolerance, lethargy, tachypnea, seizures, and coma) in the first year of life. The first description of this disorder was in 2014 by van Karnebeek et al. [1].

They reported four children from three unrelated families with acute lethargy and tachypnea associated with significant metabolic abnormalities. A male patient from the second family was of Russian origin to nonconsanguineous parents and had the homozygous synonymous single nucleotide variant NM_001739.2:c.555G>A, p.Lys185=, in the *CA5A* gene. This variant affects the last nucleotide of exon 4 and disrupts the exon 4 donor splice site, leading to in-frame skipping of exon 4. This deletion is predicted to significantly impair CAVA enzyme activity by removing amino acid residues from the active site of the CAVA enzyme or even could result in formation of protein which cannot fold properly and is likely to degrade [1]. Here, we present a clinical case of a male patient from nonconsanguineous Russian parents with the same clinical symptoms and genetic variant. We calculate the frequency of carrying this mutation in Russia.

## 2. Results

### 2.1. Clinical Characteristics

The 5-year-old male patient is the second son of healthy, nonconsanguineous parents, the older son (7 years old) is healthy. Delivery was at 40 weeks of gestation with a birth weight of 3040 g (−0.65 SDS) and length of 50 cm (0.06 SDS). The APGAR score was 8/9. At 3 days 4 h of life, the patient presented with lethargy, seizures, jaundice, respiratory distress, and intestinal bleeding. He was resuscitated with fluid boluses, given a bicarbonate correction, started on antibiotics, and transferred to the intensive care unit for ongoing management. He required mechanical ventilation for 7 days. Initial investigations showed hyperlactatemia (9.2 mmol/L), mild hypoglycemia (2.4 mmol/L), metabolic acidosis, hypernatremia, and hyperkalemia, and significant elevation of liver transaminases (ALT and AST, 10 times more than the norm), creatinine, and urea. Due to technical difficulties, the level of blood ammonia was not checked. No infection was found, and neuroinfection was excluded. The electroencephalogram (EEG) showed diffuse cerebral dysfunction and electrographic generalized seizures. Antiepileptic therapy was initiated by administering Keppra. Clinical seizures stopped on the 8th day of life. The MS/MS blood test for acylcarnitines and amino acids showed elevation of alanine and proline. The acylcarnitine level was normal. Urine analysis by gas chromatography–mass spectrometry (GC–MS) revealed significant elevation of lactate with slightly elevated pyruvate and grossly increased levels of 3-hydroxybutyric and 2-ketoglutaric acid. Pyruvate carboxylase deficiency was suspected. Analysis using an NGS-based panel that included the *PC* gene was performed. No causative variant in genes included in the panel was found. Metabolic therapy with L-carnitine and Cytoflavin (a multivitamin complex supplement containing succinic acid, nicotinamide, inosine, and riboflavin) was initiated with improvement in the patient’s condition. A repeat MS/MS blood test after 14 days showed normal acylcarnitine and amino acid profiles. However, at the age of 23 days, a respiratory obstructive syndrome began and surgical cicatricial stenosis of the larynx was performed. Subsequent motor and neurological development have remained normal: head balance from 1.5 months, sitting independently from 7 months, and walking independently at 1 year. The patient was treated with Keppra for 2 years. He is currently 5 years old, growing well, and meeting all developmental milestones without receiving additional medications. His weight is 14.5 kg (−1.58 SD) and his height is 101 cm (−1.88SD). The EEG shows multi-regional epileptiform discharges. The electrocardiogram indicates bradycardia and migration of the wandering atrial pacemaker. The level of ammonia in his blood is normal.

### 2.2. DNA Analyses

Whole genome sequencing (WGS) identified a synonymous variant c.555G>A (RefSeq NM_001739.1) in the *CA5A* gene in the homozygous state. Sanger sequencing results demonstrated variant c.555G>A in the homozygous state in the proband and in the heterozygous state in both parents and his older sibling.

### 2.3. Population Analysis

We calculated the c.555G>A (RefSeq NM_001739.1) allele frequency from the gnomAD database. It appeared to be 0.00033 (95% CI: 0.00024–0.00044) versus 0.00235 (0.0015–0.0035) from the RUSeq database. Based on the allele frequencies, the prevalence of the genetic condition was calculated to be 1:9,396,560 (1:5,165,289–1:17,361,111) and 1:180,795 (1:81,633–1:444,444), respectively. The frequency of healthy heterozygous carriers was 1:1533 (1:1137–1:2084) and 1:213 (1:143–1:334) accordingly. Based on this analysis, we conclude that the estimated prevalence of the disease according to the gnomAD database (focused on Western European populations) and the RUSeq database (based on data from the European part of Russia) differ significantly by almost 50-fold (*z*-test for two proportions, *p*-value = 4.44089 × 10^−16^).

## 3. Discussion

The CAVA enzyme supplies bicarbonate as a substrate of four mitochondrial enzymes, and deficiency of this enzyme can present with different symptoms including hyperammonemia and lactic acidosis. CAVA deficiency is an extremely rare or perhaps underdiagnosed cause of metabolic disorders, and this might cause diagnostic confusion with other inborn errors of metabolism. This disorder was first described in 2014 by van Karnebeek et al. They reported four children in three unrelated families (one consanguineous) who presented with lethargy, hyperlactatemia, and hyperammonemia. The 1st family was of Belgian–Scottish origin, the 2nd family was from Russia, and the 3rd patient was of Pakistani parents. In 2016 Carmen Diez-Fernandez et al. described 10 additional children of Turkish, Indian, Pakistani, and Bangladeshi origin [2]. In 2020, two different descriptions from India reported on five children with CAVA deficiency [3,4], and 19 affected individuals have been reported to date. Different biochemical changes can present in CAVA deficiency, but hyperammonemia, hyperlactatemia, ketonuria, and increased levels of alanine and proline in blood are always found. Only 15 pathogenic variants (as of July 2022) are included in the Human Gene Mutation Database (HGMD) v. 2022.1.

Here, we describe a boy of nonconsanguineous, healthy parents of Russian origin having a homozygous variant in the *CA5A* gene. In 2014, van Karnebeek et al. described the same pathogenic variant in the homozygous state in a 6-month-old male patient from Russia. These two patients had similar age of manifestation (on the 4th day of life) and clinical and biochemical symptoms (Table 1). However, we do not know the level of ammonia in the blood of our patient during his crisis.

Most of the previously reported patients manifested a single acute metabolic crisis and then remained normal. Only two children had learning difficulties and speech delays [1,2]. One child had multiple episodes of metabolic decompensation at 8 months and died at 22 months of age [4]. The ranges of initial presentations and long-term prognoses are still not well understood.

## 4. Materials and Methods

The family of our proband was clinically examined at the Federal Research Centre of Nutrition and Biotechnology and the Research Centre for Medical Genetics (Moscow, Russia). This study was approved by the local ethics committee of the Research Centre for Medical Genetics (approval number 2018-1/3).

### 4.1. DNA-Testing

WGS was performed using a 150-bp paired-end sequencing using the BGISEQ/MGISEQ platform. Bioinformatics analysis was performed using an in-house software pipeline designed to detect single-nucleotide variants (SNVs), copy number variations (CNVs), and mtDNA variants.

### 4.2. Population Analysis

The gnomAD v2.1.1 database (https://gnomad.broadinstitute.org/, accessed on 22 August 2022) [5] and the RUSeq database (http://ruseq.ru/, accessed on 22 August 2022) [6] were used to estimate the population allele frequency of the NM_001739.2:c.555G>A variant in the *CA5A* gene. Statistical analysis was performed using the WINPEPI software v. 11.65 [7].

## 5. Conclusions

In summary, we have analyzed CAVA deficiency as a novel differential diagnosis of neonatal and infantile hyperammonemia, and it seems more common than other rare urea cycle disorders such as N-acetylglutamate synthase (NAGS) deficiency. It is possible that other patients diagnosed with transient hyperammonemia during infancy could have been due to CAVA deficiency. This novel disease exhibits a unique biochemical profile including hyperammonemia, lactatemia and ketonuria, metabolic acidosis, hypoglycemia, and excretion of carboxylase substrates and related metabolites. Our data shows high frequency of the p.Lys185Lys (c.555G>A) variant in the *CA5A* gene in Russia, and we suggest that targeted testing of this mutation might be useful for early detection via selective screening in neonatal intensive care units.

## Figures and Tables

**Table 1 ijms-23-15026-t001:** Biochemical findings observed in our patient and the previously described patient having the same mutation [1].

Possible Enzyme Deficiency	Actual Results				
	Biomarkers	Proband 1 (Male) Clara D. van Karnebeek et.al	Normal	Proband 2 (Male) Our Patient	Normal
Carbamoyl phosphate synthetase	plasma ammonia (µmol/L)	422	<50	NA	<50
	plasma citrulline (µmol/L)	17	3–36	24	3–90
	plasma arginine (µmol/L)	35	17–119	26	0.73–90
	plasma glutamine (µmol/L)	2.606	243–822	NA	
	plasma ornithine (µmol/L)	146	38–272	127	25–700
	urine orotate	2.2	<4.9	NA	
Pyruvate carboxylase	serum glucose (mM/L)	2.9	3.0–8.0	2.4	3.0–8.0
	serum lactate (mM/L)	8.1	1.0–1.8	9.2	1.0–1.7
	plasma alanine (µmol/L)	1078	132–455	1382	95–1200
	plasma proline (µmol/L)	625	78–523	924	52–680
	urine lactate	28,000	<456	6633	<25
	urine pyruvate	NA		78.91	<12
	urine 3-OH-butryric acid	7060	<22	1413	<3
	urine aceto-acetic acid	+		NA	
	urine fumaric acid	8	<13	20.65	<2
	urine 2-oxoglutaric acid	NA		226	<152
	urine 2-a-ketoglutaric acid	300	<267	NA	
	urine adipic acid	340	<25	60	<12
	urine suberic acid	29	<15	NA	
	urine sebacic acid	NA		NA	
	plasma lysine (µmol/L)	306	71–272	NA	
Proprionyl-CoA carboxylase	urine 3-OH-propionic acid	59	<21	20.49	3–10
	urine propionylglycine	5.6	<2	NA	
	urine methylcitrate	Normal		NA	
3-methylcrotonyl-CoA carboxylase	urine 3-methylcrotonylglycine	17	<5	NA	
	urine 3-OH-isovaleric acid	327	<55	31.7	0–46

All values in urine are expressed as µmol/mmol of creatinine. +—positive, NA—not available. The values with maximal deviation from normal during crisis are provided for both probands.

## Data Availability

Not applicable.

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
