# Peer review of "Hyperammonemia in Russia Due to Carbonic Anhydrase VA Deficiency Caused by Homozygous Mutation p.Lys185Lys (c.555G>A) of the CA5A Gene"

_ijms, 2022, doi:10.3390/ijms232315026_

Round 1

Reviewer 1 Report

The authors describe a case of Carbonic anhydrase deficiency, a very rare metabolic disorder causing hyperammonemia, metabolic and lactic acidosis and ketonuria

Suggestions

Line 14 would be presenting instead of presented

line 17 should be present instead of presented

line 55 should be of instead of in

line 61 he needed instead of was needed

There is repetition between introduction and discussion which should be addressed

The initial ammonia level is missing, the lactate levels, pH and bicarb values as well as urine ketones and blood ketones should be included

What is Cytoflavin?

What is cicatricle stenosis?

Was newborn screening done?

It appears that the diagnosis was made on whole genome sequencing.  

The authors should also give the postulation of why the patients with carbonic anhydrase deficiency improve

Author Response

First of all, we eager to thank you for detailed and thorough analysis of our manuscript as well as valuable comments and recommendations to improve it. Thereunder you could find responses to each comment.

We have updated the text accordingly your suggestions.

Question 1.

The initial ammonia level is missing, the lactate levels, pH and bicarb values as well as urine ketones and blood ketones should be included

Response 1.

Thank you for the comment. Unfortunately, we don't know the initial level of ammonia in blood since it was not checked because of technical problems (they did not have ammonia-tester 6 years ago). We also do not know pH, the levels of bicarbonate and ketones in blood and urine because this information is absent in the discharge from the hospital, and we cannot get it since this information was lost.

Question 2.

What is Cytoflavin?

Response 2.

Cytoflavin is a multivitamin complex supplement which contains succinic acid, nicotinamide, inosine, and riboflavin. This clarification we included in the text (line 75-76).

Question 3.

What is cicatricle stenosis?

Response 3.

Thank you for the comment. We mean “cicatrical stenosis”. We corrected it.

Question 4.

Was newborn screening done?

Response 4.

Yes, it was done. But in Russia we have screening only for 5 conditions: PKU, congenital hypothyroidism, cystic fibrosis, classic galactosemia and congenital adrenal hyperplasia.

Question 5.

It appears that the diagnosis was made on whole genome sequencing.  

Response 5.

Yes, you are right. We performed whole genome sequencing.  

Question 6.

The authors should also give the postulation of why the patients with carbonic anhydrase deficiency improve

Response 6.

The aim of this article is to describe the clinical case and to estimate the prevalence of this rare (maybe underdiagnosed) disease in Russia. The reason for and possible causes of why the patients with carbonic anhydrase deficiency improve with time was given in details elsewhere [Diez-Fernandez C, Rüfenacht V, Santra S, Lund AM, Santer R, Lindner M, et al. Defective hepatic bicarbonate production due to carbonic anhydrase VA deficiency leads to early-onset life-threatening metabolic crisis. Genet Med. 2016;18(10):991-1000]. We have decided not to repeat it and just cited this reference.

Reviewer 2 Report

The authors present an extensive study in understanding CAVA deficiency-associated pathology and epidemiology. The study provides an in-depth biochemical profile of a 5-year-old patient's journey with the hereditary disease due to a homozygous mutation c.555G>A in the CAVA gene. The study presents the myriad of clinical characteristics that this patient poses over the course of 5-years. 

Would be great to include the Whole Genome Sequencing and Sanger sequencing data analyses as supplementary.

Minor edits in terms of sentence comprehension for:

Page 1 lines 41-42- if not lead to ...please reframe the sentence. Are you suggesting that the deletion can result in protein misfolding and subsequent degradation?

Page 2 Line 61- He was needed mechanical... should be re-written as The patient required ventilation...

Page 2 Line 71- significantly should be significant.

Page 3 Lines 73-75 - The NGS-based...among them. Please reframe the sentence to deliver the meaning clearly.

Author Response

First of all, we eager to thank you for detailed and thorough analysis of our manuscript as well as valuable comments and recommendations to improve it. Thereunder you could find responses to each comment.

We have updated the text accordingly your suggestions.

Question 1.

Would be great to include the Whole Genome Sequencing and Sanger sequencing data analyses as supplementary.

Response 1.

Although we indeed have diagnosed the patient with the aid of WGS with subsequent validation of its results by Sanger sequencing, we have found the single nucleotide variant which is previously been described elsewhere. We have decided not to include the results of these methods because here we focus on the description of the clinical case and estimation of the prevalence of this rare (maybe underdiagnosed) disease in Russia but not to description of any novel SNV.

Question 2.

Page 1 lines 41-42- if not lead to ...please reframe the sentence. Are you suggesting that the deletion can result in protein misfolding and subsequent degradation?

Response 2.

Yes. We have cited the article where the suggestion is made that the aberrant splicing could result in formation of protein which could not fold properly and is targeted to degradation. We have rephrased this sentence.

Question 3.

Page 2 Line 61- He was needed mechanical... should be re-written as The patient required ventilation...

Response 3.

We have changed the sentence accordingly.

Question 4.

Page 2 Line 71- significantly should be significant.

Response 4.

We have changed the sentence accordingly.

Question 5.

Page 3 Lines 73-75 - The NGS-based...among them. Please reframe the sentence to deliver the meaning clearly.

Response 5.

We have rephrased this sentence in order to improve its comprehension.

Round 2

Reviewer 1 Report

In Page 3 of 6 Length is recoded as 10 cm which is not accurate

line 86- rhythm driver? what does that mean

Author Response

We eager to thank you  again for detailed and careful analysis of our manuscript! We have updated the text accordingly you recommendations.
